# Unsupervised learning with spatial embedding and human labeling

## Abstract

Large-scale foundation models such as CLIP and DINOv2 provide powerful pre-trained visual embeddings that enable strong zero-shot transfer and facilitate unsupervised learning. However, for specific tasks, the visual embeddings extracted from these foundation models may still lack sufficient classification separability, making it challenging to identify a reliable classifier in the embedding space. To address this, we propose an unsupervised learning approach with spatial embedding and human labeling (SEAL). SEAL first extracts spatial embeddings using a graph attention network to capture relational cues among image patches. These spatial embeddings are then fused with foundation model features via mutual distillation, producing spatially aware representations with enhanced separability. Subsequently, a lightweight linear classifier is trained in this embedding space to generate cluster assignments that reflect human labeling. Experimental analysis on 26 benchmark datasets shows that incorporating spatial embeddings significantly improves triplet accuracy, demonstrating the enhanced separability of foundation model embeddings. Extensive experiments further show that SEAL achieves outstanding clustering performance across 26 benchmark datasets and maintains excellent stability across 7 foundation model backbones. The code will be released publicly.

## 1 Introduction

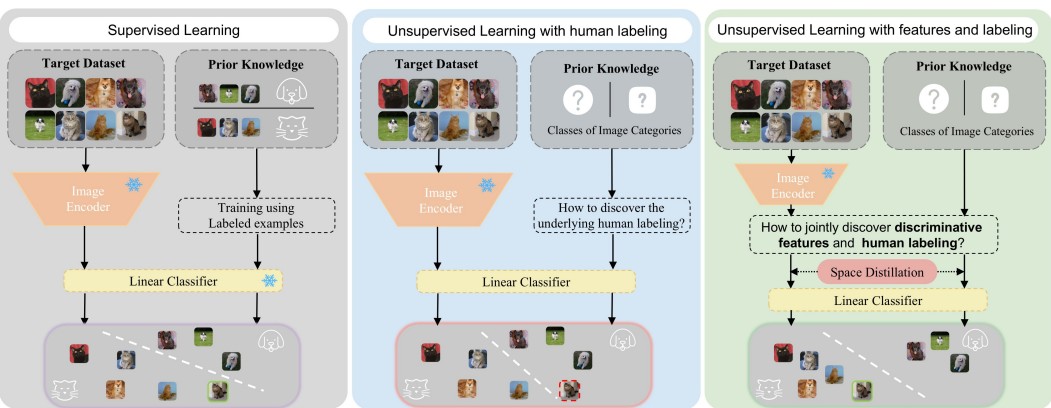

Figure 1: Comparison of Supervised and Unsupervised Learning Approaches. (a) Supervised learning trains models on labeled examples with predefined image categories. (b) Unsupervised learning infers underlying human labeling without labeled examples, relying only on the specified number of categories. (c) Unsupervised learning simultaneously discovers discriminative features and recovers human labeling, only given the number of categories.

Deep learning has revolutionized computer vision by enabling models to learn representations with rich information directly from raw images (LeCun et al., 2015). In supervised learning (illustrated on the left of Figure 1), labeled datasets provide the critical training signal (Krizhevsky et al., 2012;

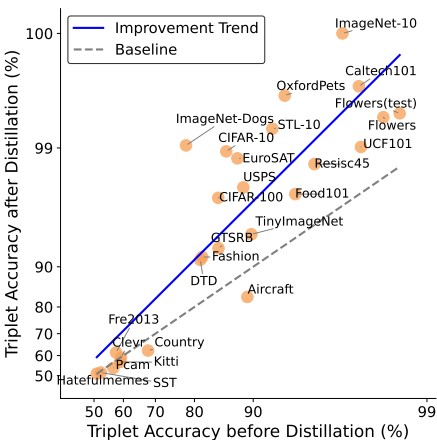 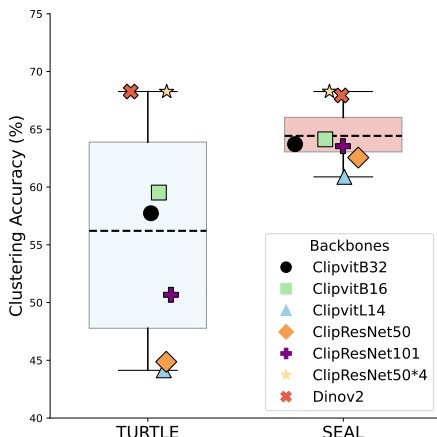

(a) Triplet accuracy comparison between different datasets with and without spatial structure using ClipVIT-L/14. The blue line shows the overall improvement trend, and the gray dashed line indicates similar triplet accuracy before and after distillation.

(b) Boxplots depict the distribution of the average clustering accuracies on 26 datasets for TURTLE (blue) and SEAL (red) across 7 backbones. Individual points indicate the average accuracy achieved by each backbone.

Figure 2: (a) Triplet accuracy before and after distillation. (b) Clustering performance across backbones.

He et al., 2016; Ren et al., 2015), and the learned representations serve as the foundation for achieving excellent performance on core tasks such as image classification, object detection, and semantic segmentation. In parallel, unsupervised representation learning aims to uncover meaningful information from raw images without relying on human annotations. Early representative methods in this field, such as Deep Embedded Clustering (DEC) (Xie et al., 2016) and Contrastive Clustering (CC) (Li et al., 2021), have shown that leveraging the intrinsic properties of the data enables models to learn transferable embeddings that generalize effectively across downstream tasks.

With the accumulation of data across multiple domains, numerous large-scale foundation models have emerged, including CLIP (Radford et al., 2021), DINOv2 (Oquab et al., 2023), SWAG (Singh et al., 2022), and CoCa (Yu et al., 2022). Their pre-trained visual embeddings have evolved into powerful priors for a wide range of downstream tasks, exhibiting strong zero-shot transfer capabilities by aligning input data with human-provided prompts. However, reliance on such external supervision limits their applicability in fully unsupervised scenarios. To address this limitation, recent methods such as HUME (Gadetsky & Brbic, 2023) and TURTLE (Gadetsky et al., 2024) reinterpret clustering as the task of discovering human-consistent labelings from pre-trained embeddings. The core insight behind the methods is that human-defined categories tend to be linearly separable in the representation space. This property allows for searching through potential labelings and evaluating them based on the performance of linear classifiers. Using this principle, these methods aim to recover semantically meaningful data partitions without relying on human-provided prompts, thereby advancing toward fully unsupervised utilization of foundation models.

However, for specific tasks, the pre-trained embeddings, despite containing rich semantic information, may still lack sufficient linear separability, making it challenging to identify a reliable linear classifier in the feature space.

To address this limitation, we propose a method to enhance the linear separability of visual embeddings extracted from foundation models. Our approach starts with a Graph Attention Network (GAT) encoder, which extracts spatial embeddings by capturing relational cues between image patches. These spatial embeddings are then fused with the visual embeddings of the foundation via mutual distillation to generate spatially aware embeddings that better retain the structural information of images. To quantify the improvement of the spatially aware embeddings in separability, we adopt a triplet accuracy, which evaluates the relative closeness of samples from the same class versus those from different classes, thereby reflecting the linear separability of features (see Appendix D for details). As illustrated in Figure 2a, integrating spatial structure into foundation model embeddings

yields significantly higher triplet accuracy, showing that the linear separability of features is effectively enhanced. We then discover actual clustering assignments for the data based on the spatially aware embeddings.

Building on the above, we introduce an unsupervised learning method with spatial embedding and human labeling (SEAL), which integrates spatial structures to enhance feature separability and recover the underlying human labeling simultaneously. Following the process detailed earlier (i.e., spatial embedding extraction via GAT and mutual distillation with visual embeddings from foundation models), inspired by (Gadetsky et al., 2024), a lightweight linear classifier is trained on this spatially aware embedding space to output cluster assignments. We evaluated the performance of SEAL on 26 benchmark datasets and 7 foundation model backbones (including CLIP ResNets, CLIP Vision Transformers, and DINOv2). As illustrated in Figure 2b, SEAL outperforms the TURTLE baseline. In particular, it achieves more stable performance across diverse foundation model backbones. Detailed results for each dataset are provided in Appendix C. In addition, although spatial embedding consumes extra time, the clearer class structure improves the efficiency of recovering human labeling, and the efficiency of SEAL is still at a comparable level. These results confirm that explicitly modeling spatial structures provides dual benefits: it enhances the linear separability of features (as validated by triplet accuracy) and improves the effectiveness and stability of clustering, thereby advancing the state-of-the-art in unsupervised learning.

The main contributions are as follows:

- We propose a spatially aware approach that generates spatial features and undergoes mutual distillation with embeddings from foundation models, thereby capturing spatial structure relationships and enhancing separability. The improvement in separability is quantitatively demonstrated across 26 datasets and 7 backbones using triplet accuracy.
- We propose an unsupervised learning method with spatial embeddings and human labeling (SEAL), which recovers human labeling based on spatially aware embeddings. Extensive experiments demonstrate that SEAL achieves outstanding clustering performance on 26 benchmark datasets and exhibits excellent stability across 7 foundation model backbones.

## 2 RELATED WORK

### 2.1 REPRESENTATION LEARNING FOR IMAGE CLUSTERING

Classical methods such as K-means or spectral clustering (Krishna & Murty, 1999; Ng et al., 2001) rely on low-level embeddings and often fail when applied to large-scale datasets with complex semantics.

With the advent of deep learning, image clustering performance has significantly improved by leveraging neural networks. These methods rely on internal supervision signals to guide the learning of clustering-friendly representations. They typically assume that samples from the same category are naturally closer in the embedding space and optimize the representations accordingly. Early works like Deep Embedded Clustering (DEC) (Xie et al., 2016) refine cluster assignments by minimizing the KL divergence between current predictions and a sharpened target distribution. Prediction-consistency based approaches such as Invariant Information Clustering (IIC) (Ji et al., 2019), SCAN (Van Gansbeke et al., 2020), and Contrastive Clustering (CC) (Li et al., 2021) leverage agreement under strong data augmentations to facilitate unsupervised clustering. Alternatively, iterative pseudo-labeling methods, including DeepCluster (Caron et al., 2018) and SPICE (Niu et al., 2022), repeatedly assign cluster labels to supervise representation learning. Graph-based extensions like GATCluster (Niu et al., 2020) introduce graph attention mechanisms to model neighborhood dependencies and improve clustering quality.

### 2.2 UNSUPERVISED LEARNING WITH FOUNDATION MODELS

While deep clustering methods learn task-specific embeddings, recent works leverage large-scale pretrained models to provide rich visual representations, with CLIP and CoCa capturing high-level semantic concepts through image–language alignment, and DINOv2 and SWAG excelling at preserving visual details. While these models demonstrate strong zero-shot transfer, their reliance on external supervision limits applicability in fully unsupervised settings.

To address this, recent works such as HUME (Gadetsky & Brbic, 2023) and TURTLE (Gadetsky et al., 2024) reframe clustering as the task of discovering human labeling directly from pretrained embeddings. HUME introduced the idea of searching for labelings that are linearly separable across multiple representation spaces, using self-supervised features for the task encoder and a large pre-trained model as a regularizer. However, it still required task-specific self-supervised pretraining on the target dataset. TURTLE advances this paradigm by entirely removing the need for dataset-specific representation learning, instead leveraging fixed embeddings from one or two foundation models to define the task encoder and evaluate generalization error. By maximizing the margin of linear classifiers across these spaces, TURTLE effectively identifies labelings that closely align with human labeling, often matching or surpassing zero-shot transfer methods' performance without any human labeling.

These developments highlight a broader shift in unsupervised learning: moving from learning embeddings from scratch to leveraging pretrained foundation models to recover semantically meaningful data partitions. This trend improves scalability and efficiency, and narrows the gap between fully unsupervised and supervised learning. Nevertheless, the lack of sufficient class separability remains a persistent challenge in unsupervised learning.

## 3 THE PROPOSED METHOD

### 3.1 PROBLEM STATEMENT AND OBJECTIVE

In this paper, we consider the following unsupervised learning problem: given an unlabeled training dataset $\mathcal{D} = \{x_n\}_{n=1}^N$ with a known number of categories $K$, the goal is to learn a model $f : \mathcal{D} \to \{1, \ldots, K\}$ that assigns a pseudo-label $y_i = f(x_i)$ to each sample $x_i$, such that samples belonging to the same class exhibit homogeneity.

We address this problem through spatial embedding and human labeling. Each image $x$ is represented by a pre-trained visual embedding $\phi(x) \in \mathbb{R}^d$. Our approach has two main objectives: (i) to enhance the linear separability of the visual embeddings, and (ii) to infer a human-consistent labeling distribution $\tau_\theta(x) \in \Delta^{K-1}$, which assigns semantically meaningful category probabilities to each sample, where $\Delta^{K-1}$ denotes the $(K-1)$-dimensional probability simplex. Formally, we leverage additional spatial embeddings $g(x) \in \mathbb{R}^d$. By integrating these spatial embeddings with the visual embeddings $\phi(x)$, we aim to produce spatially aware embeddings $\hat{\phi}(x)$ that improve linear separability and make clustering categories more distinguishable in the feature space. Subsequently, we learn a mapping $\hat{\phi}(x) \in \mathbb{R}^d$ such that a linear classifier $f(\hat{\phi}(x)) = w^\top \hat{\phi}(x)$ assigns samples to the $K$ clusters according to the inferred distribution $\tau_\theta(x)$. An overview of the proposed framework is illustrated in Figure 3.

Guided by the above solution, we propose a learnable distillation function $\mathcal{R}$ that integrates $\phi(x)$ and $g(x)$ to obtain spatially-aware embeddings $\hat{\phi}(x) = \mathcal{R}(\phi(x), g(x))$. A classifier $f(x) = w^\top \hat{\phi}(x)$ is then trained on this embedding space. The overall optimization problem is formulated as:

$$\min_\theta \quad \mathcal{L}(\theta) = \sum_{x \in \mathcal{D}} \mathcal{L}_{ce}\big(f(x), \tau_\theta(x)\big) - \beta \mathcal{L}_{ent}(\tau_\theta(x)), \tag{1}$$

$$\text{s.t.} \quad \left\{ \begin{array}{l} \hat{\phi}(x) = \mathcal{R}(\phi(x), g(x)), \\ \mathcal{R} = \arg\min_{\mathcal{R}'} \mathcal{L}_{\text{distill}}\big(\phi(x), g(x); \mathcal{R}'\big) \\ f(x) = w^\top \hat{\phi}(x), \\ w = \arg\min_{w'} \mathcal{L}_{\text{ce}}\big(w'^\top \hat{\phi}(x), \tau_\theta(x); w'\big) \end{array} \right\} \begin{array}{l} \text{spatially aware embedding} \\ \\ \text{human labeling} \end{array} \tag{2}$$

where $\beta$ is a weighting factor for the entropy loss of the latent label distribution, and $\mathcal{L}_{\text{ent}}^{\tau_\theta}$ denotes the entropy loss of $\tau_\theta(x)$. The spatial embedding $g(x)$ is obtained as described in Section 3.2 and the learning of $\mathcal{R}$ is detailed in Section 3.3. The human labeling $f(x)$ on spatially aware embeddings is introduced in Section 3.4. The entire framework is trained by jointly optimizing $\theta$, $\mathcal{R}$, and $w$.

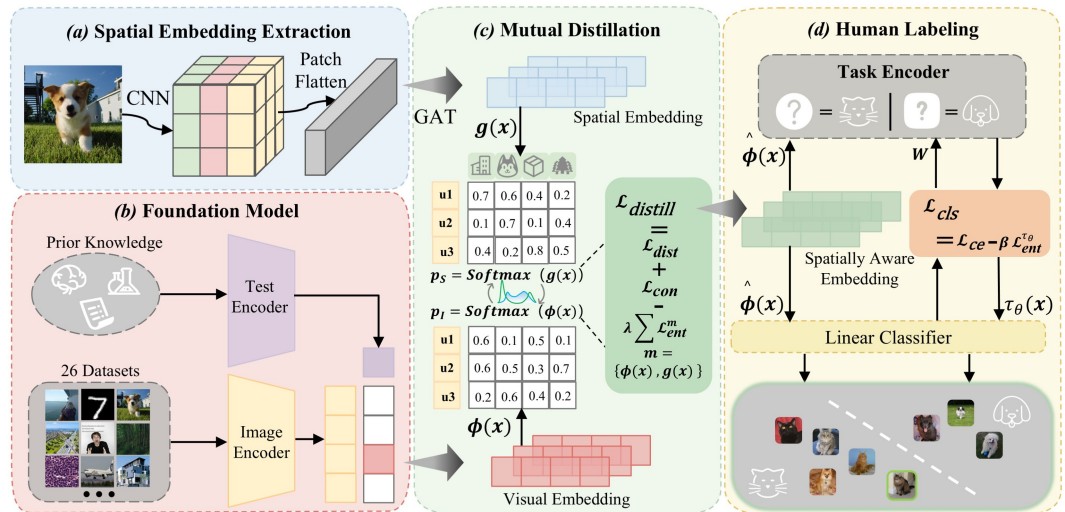

Figure 3: Overview of the proposed SEAL framework.

## 3.2 SPATIAL EMBEDDING EXTRACTION

Capturing spatial structure is crucial for distinguishing task-specific categories. Inspired by the structural representation idea in (Qian et al., 2015), we model spatial dependencies among image patches using GAT (Veličković et al., 2017). Since raw image dataset $\mathcal{D}$ cannot be directly input to GAT, We first feed $\mathcal{D}$ into a pretrained ResNet-50 (Radford et al., 2021) to extract high-level convolutional features, which are then used as inputs for the GAT, including node embeddings $X = [x_1, \ldots, x_N]^\top \in \mathbb{R}^{N \times d}$ and edge information $E = \{(i,j) \mid i \neq j, \ i,j \in \{1, \ldots, N\}\}$.

Then, $X$ and $E$ are input to the GAT, which updates each node by attending to its neighbors:

$$x_i' = \sum_{j \in \mathcal{N}(i)} \alpha_{ij} W x_j, \quad \alpha_{ij} = \frac{\exp\left(\text{LeakyReLU}\left(a^\top [W x_i \parallel W x_j]\right)\right)}{\sum_{k \in \mathcal{N}(i)} \exp\left(\text{LeakyReLU}\left(a^\top [W x_i \parallel W x_k]\right)\right)}, \quad (3)$$

where $W$ is a learnable weight matrix, $\mathcal{N}(i)$ denotes the neighbors of node $i$, $a$ is a learnable vector, and $\parallel$ denotes vector concatenation.

After refining the node embeddings, we apply global average pooling to $X' = [x_1', \ldots, x_N']^\top \in \mathbb{R}^{N \times d}$ in order to generate the spatial embedding:

$$g(x) = \frac{1}{N} \sum_{i=1}^{N} x_i' \in \mathbb{R}^d. \quad (4)$$

In the following section, we fuse the spatial embedding $g(x)$ derived above with the visual embedding $\phi(x)$ to obtain spatially aware embeddings.

## 3.3 MUTUAL DISTILLATION

Knowledge distillation (KD) (Hinton et al., 2015) has been shown to enhance embedding consistency through cross-model alignment (Radford et al., 2021). Motivated by this, we introduce a mutual distillation framework to obtain spatially aware embeddings by jointly leveraging both visual and spatial modalities.

Given an input image $x$, we first extract visual features $\phi(x)$ using the foundation model and obtain spatial features $g(x)$ as described in the previous section. The logits for the visual and spatial modalities are then computed via their respective clustering heads:

$$\text{logit\_I} = h_I(\phi(x)), \quad \text{logit\_S} = h_S(g(x)), \quad (5)$$

where $h_I$ and $h_S$ denote MLP projection heads for visual and spatial modalities, respectively. The spatially aware embeddings are subsequently obtained by integrating the two modalities through a learnable distillation function (corresponding to the spatially aware embedding component in Eq. 2):

$$\hat{\phi}(x) = \mathcal{R}(\phi(x), g(x)), \quad \mathcal{R} = \arg\min_{\mathcal{R}'} \mathcal{L}_{\text{distill}}\big(\phi(x), g(x); \mathcal{R}'\big). \tag{6}$$

The optimization objective $\mathcal{L}_{distill}$ combines three terms, which are distillation loss, consistency loss, and entropy loss. The mutual distillation loss aligns the cluster distributions between visual and spatial modalities:

$$\mathcal{L}_{\text{dist}} = \mathcal{L}_{S\to I} + \mathcal{L}_{I\to S}, \tag{7}$$

$$\mathcal{L}_{S\to I} = -\frac{1}{N}\sum_{i=1}^{N} \log \frac{e^{\text{sim}(\text{logit\_I}_i,\text{logit\_S}_i)/T}}{\sum_{j=1}^{N} e^{\text{sim}(\text{logit\_I}_i,\text{logit\_S}_j)/T}}, \tag{8}$$

$$\mathcal{L}_{I\to S} = -\frac{1}{N}\sum_{i=1}^{N} \log \frac{e^{\text{sim}(\text{logit\_S}_i,\text{logit\_I}_i)/T}}{\sum_{j=1}^{N} e^{\text{sim}(\text{logit\_S}_i,\text{logit\_I}_j)/T}}, \tag{9}$$

where $T$ is a temperature parameter. The consistency loss enforces agreement between modalities:

$$\mathcal{L}_{\text{con}} = -\frac{1}{N}\sum_{i=1}^{N} \log\left(\text{logit\_I}_i^{\top}, \text{logit\_S}_i\right). \tag{10}$$

The entropy loss prevents degenerate solutions:

$$\mathcal{L}_{\text{ent}}^m = -\frac{1}{N}\sum_{i=1}^{N}\frac{1}{K}\sum_{j=1}^{K}(\text{logit}_{i,j}^m)\log(\text{logit}_{i,j}^m), \quad m \in \{\phi(x), g(x)\}. \tag{11}$$

The overall training objective is defined as:

$$\mathcal{L}_{distill} = \mathcal{L}_{\text{dist}} + \mathcal{L}_{\text{con}} - \lambda\sum_{m}\mathcal{L}_{\text{ent}}^m. \tag{12}$$

where the entropy regularization weight is fixed to $\lambda = 5$.

After the mutual distillation, we obtain the spatially aware embeddings from the distilled image clustering head. These embeddings preserve the spatial structural relationships of the images. In the next subsection, we leverage the spatially aware embedding space to discover the underlying human labeling, thereby generating the cluster assignment.

### 3.4 HUMAN LABELING

After obtaining the spatially aware embeddings $\hat{\phi}(x)$ via mutual distillation, we freeze them and adopt the bi-level optimization protocol of TURTLE (Gadetsky et al., 2024) to train a final linear classifier.

Specifically, given the pseudo-label distribution $\tau_\theta(x) \in \Delta^{K-1}$ produced by the task encoder at the $t$-th outer iteration, the linear classifier is updated in the inner loop on the current mini-batch $\mathcal{B}^{(t)}$ via 10 steps of SGD:

$$w = \arg\min_{w'} \mathcal{L}_{\text{ce}}\big(w'^{\top}\hat{\phi}(x), \tau_\theta(x); w'\big), \quad f(x) = w^{(t)\top}\hat{\phi}(x) \in \mathbb{R}^K. \tag{13}$$

In the outer loop, the task encoder parameters $\theta$ are updated by minimizing $\mathcal{L}_{\text{ce}}$ with an additional entropy regularization term:

$$\min_{\theta} \quad \mathcal{L}(\theta) = \sum_{x\in\mathcal{D}} \mathcal{L}_{\text{ce}}\big(f(x), \tau_\theta(x)\big) - \beta\mathcal{L}_{\text{ent}}(\tau_\theta(x)), \tag{14}$$

where the entropy regularization weight is fixed to $\beta = 10$.

Following TURTLE, the linear classifier can operate on spatially-aware embeddings generated from either one or two backbones, which we denote as 1-space and 2-space, respectively. For the 2-space case, the per-space $\tau_\theta(x)$ are first computed and then averaged to obtain the final label assignments.

# 4 EXPERIMENTS

## 4.1 EXPERIMENTAL SETTINGS

**Datasets and evaluation metric.** The experimental analysis is conducted on 26 benchmark datasets spanning object recognition, scene understanding, fine-grained classification, and remote sensing. The detailed description of each dataset is provided in Appendix A.1. Train and test splits follow the original dataset configurations. To evaluate the performance, we adopted Accuracy (ACC) (Xie et al., 2016), Normalized Mutual Information (NMI) (Estévez et al., 2009) and Adjusted Rand Index (ARI) (Steinley, 2004). All experiments are repeated ten times with different random seeds, and the average indices values are reported.

**Baselines.** We employ a range of state-of-the-art clustering methods as reference, including K-Means (Krishna & Murty, 1999), DEC (Xie et al., 2016), DAC (Chang et al., 2017), DSEC (Chang et al., 2018), DCCM (Wu et al., 2019), MICE (Tsai et al., 2020), GATCluster (Niu et al., 2020), PLCA (Huang et al., 2020), IDFD (Tao et al., 2021), CC (Li et al., 2021), C3 (Sadeghi et al., 2022), TCL (Li et al., 2022), CONCUR (Deshmukh et al., 2022), SPICE (Niu et al., 2022), TAC (Li et al., 2023), DPAC (Yan et al., 2024), and TURTLE (Gadetsky et al., 2024).

**Foundation Representations.** We employ CLIP (Radford et al., 2021) representations across different architectures and model sizes, including three ResNets (ResNet50, ResNet101, ResNet50x4), three Vision Transformers (VIT-B/32, VIT-B/16, and VIT-L/14), and DINOv2 VIT-G/14 (Oquab et al., 2023). For clarity, we refer to SEAL using a representation space generated from a single backbone as 1-Space, and SEAL combining two representation spaces as 2-Space. The specific backbones employed in each experiment are indicated in the corresponding sections. Additional details on the models and the procedure for preparing representations are provided in Appendix A.2.

**Implementation Details.** We implement Mutual Distillation with two MLP-based cluster heads. For the features of one modality, we select the top-50 nearest neighbors from the other modality's features to calculate loss. The cluster heads are trained with Adam ($lr = 0.001, \gamma = (0.9, 0.99)$) for 20 epochs (batch size 512) and temperature $t = 1.0$. Input embeddings have backbone-dependent dimensions, and the final spatially aware embeddings retain the same dimensionality as the input. We implement linear classifier using a bi-level optimization scheme. The outer loop runs for $T = 6000$ iterations with Adam (outer_lr $= 0.001, \gamma = (0.9, 0.999)$) and batch size 10,000, updating the task encoder. The inner loop performs $M = 10$ steps per outer iteration with Adam (inner_lr $= 0.001, \gamma = (0.9, 0.999)$), updating the weight parameters $w$ of the linear classifier.

## 4.2 EXPERIMENTAL RESULTS

### 4.2.1 COMPARATIVE ANALYSIS RESULTS

For clarity, in this comparison, TURTLE (1-space) and SEAL (1-space) both use features extracted from the DINOv2 backbone, whereas TURTLE (2-space) and SEAL (2-space) leverage dual-space features from CLIP VIT-L/14 and DINOv2 backbones.

The results on 4 widely used benchmark datasets are reported in Table 1. SEAL achieves state-of-the-art performance on ImageNet-10, ImageNet-Dogs, and STL-10, with nearly perfect clustering results on ImageNet-10 and STL-10 datasets. On CIFAR-10, SEAL performs competitively and remains close to TURTLE (Gadetsky et al., 2024), leading to the best overall average ACC performance across all methods. In this comparison, the performance of SEAL (1-space) is superior to that of TURTLE (1-space). More comparison experiments between SEAL, K-Means, and TURTLE can be found in Appendices E and F. These results illustrate the effectiveness of SEAL.

### 4.2.2 TIME ANALYSIS

We evaluate the efficiency of our method versus TURTLE on four datasets. Figure 4 shows total runtimes for increasing spatial structure and recovering human labeling using three backbones: CLIP VIT-L/14, DINOv2, and CLIP VIT-L/14 + DINOv2. Although increasing spatial structure adds some overhead, it clarifies class structure and accelerates human labeling recovery, making SEAL's overall efficiency comparable to TURTLE; in some cases, the speedup even offsets the extra time.

Table 1: Clustering performance of different approaches evaluated by ACC%, NMI%, and ARI%. "–" indicates unavailable results. **Bold** numbers indicate the best performance across all methods. Underlined numbers indicate the best performance when comparing between TURTLE (1-space) vs SEAL (1-space) and TURTLE (2-space) vs SEAL (2-space), respectively.

| Method | ImageNet-10 | | | ImageNet-Dogs | | | STL-10 | | | CIFAR-10 | | | Avg. |
|---|---|---|---|---|---|---|---|---|---|---|---|---|---|
| | ACC | NMI | ARI | ACC | NMI | ARI | ACC | NMI | ARI | ACC | NMI | ARI | |
| K-Means (Krishna & Murty, 1999) | 24.1 | 11.9 | 5.7 | 10.5 | 5.5 | 2.0 | 19.2 | 12.5 | 6.1 | 22.9 | 8.7 | 4.9 | 19.2 |
| DEC (Xie et al., 2016) | 38.1 | 28.2 | 20.3 | 19.5 | 12.2 | 7.9 | 35.9 | 27.6 | 18.6 | 30.1 | 25.0 | 16.1 | 30.9 |
| DAC (Chang et al., 2017) | 52.7 | 39.4 | 30.2 | 27.5 | 21.9 | 11.1 | 47.0 | 36.6 | 25.7 | 52.2 | 40.0 | 30.1 | 44.9 |
| DSEC (Chang et al., 2018) | 67.4 | 58.3 | 52.2 | 26.4 | 23.6 | 12.4 | 48.2 | 40.3 | 28.6 | 47.8 | 43.8 | 34.0 | 47.5 |
| DCCM (Wu et al., 2019) | 71.0 | 60.8 | 55.5 | 38.3 | 32.1 | 18.2 | 48.2 | 37.6 | 26.2 | 62.3 | 49.6 | 40.8 | 54.9 |
| MiCE (Tsai et al., 2020) | – | – | – | 43.9 | 42.3 | 28.6 | 75.2 | 63.5 | 57.5 | 83.5 | 73.7 | 69.8 | 67.5 |
| GATCluster (Niu et al., 2020) | 76.2 | 60.9 | 57.2 | 33.3 | 32.2 | 20.0 | 58.3 | 44.6 | 36.3 | 61.0 | 47.5 | 40.2 | 57.2 |
| PICA (Huang et al., 2020) | 87.0 | 80.2 | 76.1 | 35.2 | 35.2 | 20.1 | 71.3 | 61.1 | 53.1 | 69.6 | 59.1 | 51.2 | 65.8 |
| IDFD (Tao et al., 2021) | 95.4 | 89.8 | 90.1 | 59.1 | 54.6 | 41.3 | 75.6 | 64.3 | 57.5 | 81.5 | 71.1 | 66.3 | 77.9 |
| CC (Li et al., 2021) | 89.3 | 85.9 | 82.2 | 42.9 | 44.5 | 27.4 | 85.0 | 76.4 | 72.6 | 79.0 | 70.5 | 63.7 | 74.1 |
| C3 (Sadeghi et al., 2022) | 94.2 | 90.5 | 86.1 | 43.4 | 44.8 | 28.0 | – | – | – | 83.8 | 74.8 | 70.7 | 73.8 |
| TCL (Li et al., 2022) | 89.5 | 87.5 | 83.7 | 64.4 | 62.3 | 51.6 | 86.8 | 79.9 | 75.7 | 88.7 | 81.9 | 78.0 | 82.4 |
| ConCUR (Deshmukh et al., 2022) | 95.8 | 90.7 | 90.9 | 69.5 | 63.0 | 53.1 | 74.9 | 63.6 | 56.6 | 84.6 | 76.2 | 71.5 | 81.2 |
| SPICE (Niu et al., 2022) | 96.9 | 92.7 | 93.3 | 67.5 | 62.7 | 52.6 | 92.9 | 86.0 | 86.5 | 91.8 | 85.0 | 83.6 | 87.3 |
| TAC (Li et al., 2023) | 99.4 | 98.5 | 98.8 | 84.4 | 77.4 | 72.0 | 98.3 | 95.7 | 96.3 | 92.2 | 83.7 | 83.6 | 93.6 |
| DPAC (Yan et al., 2024) | 97.0 | 92.5 | 93.5 | 72.6 | 66.7 | 59.8 | 93.4 | 86.3 | 86.1 | 93.4 | 87.0 | 86.6 | 89.1 |
| TURTLE(1-space)(Gadetsky et al., 2024) | 89.1 | 88.5 | 83.0 | 87.0 | 84.8 | 78.7 | 56.2 | 58.0 | 41.3 | 90.8 | 88.4 | 83.8 | 80.8 |
| SEAL(1-space) | 99.7 | 99.1 | 99.3 | 97.8 | 95.1 | 95.4 | 87.1 | 86.6 | 78.7 | 98.6 | 96.1 | 96.8 | 95.8 |
| TURTLE(2-space)(Gadetsky et al., 2024) | 99.8 | 99.3 | 99.5 | 91.5 | 88.9 | 85.5 | 99.8 | 99.4 | 99.5 | 99.5 | 98.3 | 98.8 | 97.6 |
| SEAL(2-space) | **99.9** | **99.6** | **99.7** | 97.9 | 95.3 | 95.5 | **99.9** | **99.8** | **99.9** | 98.7 | 96.5 | 97.2 | **99.1** |

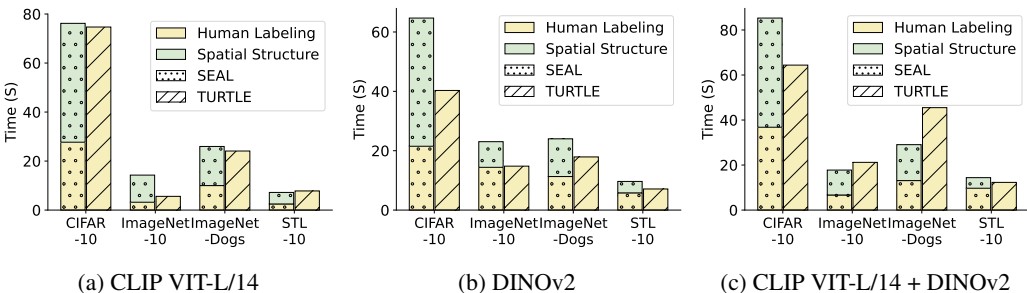

| (a) CLIP VIT-L/14 | (b) DINOv2 | (c) CLIP VIT-L/14 + DINOv2 |

Figure 4: Time consumption of SEAL and TURTLE across 4 datasets under 3 backbones.

## 4.3 Effectiveness of Spatially Aware Embeddings

To evaluate the contribution of spatially aware embeddings to representation learning, we conduct an ablation study using K-Means clustering. Specifically, we compare two settings: (1) directly concatenating the original CLIP VIT-L/14 and DINOv2 embeddings (baseline), and (2) first applying the mutual distillation to both embeddings before concatenation. We

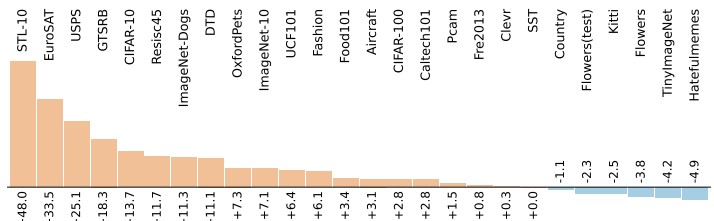

Figure 5: K-Means accuracy improvement (%) from concatenating embeddings of VIT-L/14 and DINOv2 after applying a spatially-aware transformation to each, compared to directly concatenating their original embeddings, across 26 benchmark datasets.

employ the standard K-Means algorithm, with the number of clusters set equal to the ground-truth class count for each dataset. To reduce randomness, K-Means is repeated 30 times with different centroid initializations, using a fixed random seed to ensure reproducibility. Figure 5 presents the accuracy improvements across 26 benchmark datasets. As shown, spatially aware embeddings substantially improve clustering performance on most datasets. Detailed results for each dataset are provided in Table 7 in Appendix F.

### 4.3.1 EFFECT OF MUTUAL DISTILLATION

We first evaluate the effect of Mutual Distillation (MD) on 1-space representation. Figure 6 reports the average ACC of SEAL (with MD) compared to the baseline TURTLE (without MD) across 7 representations, averaged over 26 datasets. As shown in Figure 6, SEAL consistently outperforms TURTLE in all representations. The improvement is particularly pronounced for convolutional backbones: CLIP ResNet50, CLIP ResNet101, and CLIP ResNet50*4 achieve gains of +16.8%, +17.7%, and +12.9%, respectively. Transformer-based representations (CLIP VIT-B/32, CLIP VIT-B/16, CLIP VIT-L/14) exhibit more moderate gains of +6.0%, +4.6%, and +0.01%, while DI-NOv2 shows a +3.8% improvement. These results demonstrate that Mutual Distillation enhances embedding representations. Detailed results for each dataset are provided in Appendix B.1, which are consistent with the trends

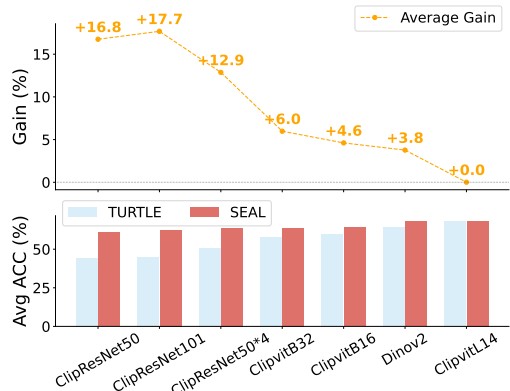

Figure 6: Effect of Mutual Distillation on 1-space representations across 26 data datasets. SEAL leverages Mutual Distillation, while TURTLE serves as the counterpart without it.

in Figure 6. Furthermore, we evaluated the average ACC performance of 2-space representations, which are provided in the Appendix B.2. The results show the enhancement of the embedding by distillation.

### 4.3.2 PARAMETER ANALYSIS

We conduct a parameter analysis of SEAL under CLIP VIT-L/14 on two representative datasets (ImageNet-10 and CIFAR-10). We vary the entropy weight ($\{4.0, 4.5, 5.0, 5.5, 6.0, 6.5\}$) and the consistency weight ($\{0, 0.5, 1.0, 1.5, 2.0, 2.5\}$), while keeping all other hyperparameters fixed. As shown in Figure 7, ACC remains stable in a wide range of parameter values. Moderate entropy regularization, combined with a consistency weight of around 1.0, generally yields good performance, demonstrating that SEAL is robust to hyperparameter variations and does not require fine-tuning.

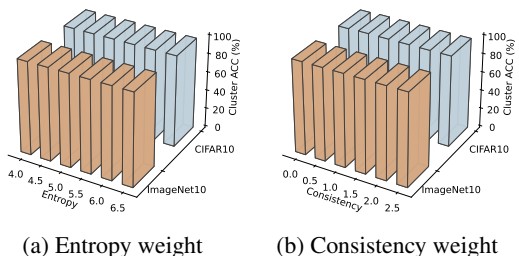

(a) Entropy weight    (b) Consistency weight

Figure 7: 3D bar plots of clustering accuracy under different entropy and consistency weights.

## 5 CONCLUSION

In this work, we addressed the challenge of limited linear separability in foundation model embeddings for unsupervised learning. We proposed SEAL, a framework that enhances feature separability by extracting spatial embeddings with a Graph Attention Network and integrating them with foundation model features through mutual distillation. Within this spatially aware embedding space, a linear classifier is trained to generate cluster assignments that better align with human labeling. The spatially aware embeddings improve both the structural fidelity and discriminability of representations, as evidenced by significantly higher triplet accuracy on 26 benchmark datasets. Extensive experiments on 26 datasets and 7 backbones further demonstrate that SEAL consistently achieves superior clustering performance and exhibits remarkable stability compared to strong baselines such as TURTLE. Although SEAL introduces additional computation, it improves recovering labeling efficiency by revealing clearer class structures. These results highlight the importance of incorporating spatial structures into foundation model embeddings to advance unsupervised visual representation learning. In future work, we plan to extend SEAL to multi-modal and large-scale streaming settings, further improving efficiency and adaptability in real-world applications.

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

APPENDIX

This appendix provides additional details for the ICLR 2026 submission, titled "Unsupervised learning with spatial embedding and human labeling". The appendix is organized as follows:

## A  EXPERIMENTAL DETAILS

### A.1  DATASETS

We evaluate our framework on 26 benchmark datasets, covering a wide range of vision tasks. These include general object classification datasets CIFAR-10 (DeVries & Taylor, 2017), CIFAR-100 (DeVries & Taylor, 2017), STL-10 (Coates et al., 2011), TinyImageNet (Le & Yang, 2015), ImageNet-10 (Chang et al., 2017), and Caltech101 (Fei-Fei et al., 2004); fine-grained object classification datasets Food101 (Bossard et al., 2014), Flowers (Nilsback & Zisserman, 2008), Flowers(Test) (Nilsback & Zisserman, 2008), FGVC Aircraft (Maji et al., 2013), ImageNet-Dogs (Chang et al., 2017), and OxfordPets (Parkhi et al., 2012); grayscale image datasets USPS (Sankaranarayanan et al., 2018), and Fashion-MNIST (Xiao et al., 2017); the texture dataset DTD (Cimpoi et al., 2014); the facial emotion recognition dataset FER2013 (Goodfellow et al., 2013); the satellite image classification datasets EuroSAT (Helber et al., 2019) and RESISC45 (Cheng et al., 2017); the German Traffic Sign Recognition Benchmark (GTSRB) (Stallkamp et al., 2012); the KITTI Distance dataset (Geiger et al., 2012); the metastatic tissue classification dataset PatchCamelyon (PCam) (Veeling et al., 2018); the CLEVR counting dataset (Johnson et al., 2017); the video dataset UCF101 (Soomro et al., 2012); the multimodal HatefulMemes dataset (Kiela et al., 2020); the country classification dataset Country211 (Radford et al., 2021); and the Rendered SST2 dataset (Radford et al., 2021).

For CLEVR, we randomly sample 2,000 train and 500 test images. For UCF101, we take the middle frame of each video clip as the input. Details of each dataset are summarized in Table 2. Finally, it is worth noting that SEAL can also be applied to tasks in various modalities beyond vision, and even to cross-modal scenarios, provided that pre-trained representations are available.

Table 2: Summary statistics of the 26 datasets, including number of classes, train size, and test size.

| Dataset | Num Classes | Train Size | Test Size |
|---|---|---|---|
| CLEVR (Johnson et al., 2017) | 8 | 2,000 | 500 |
| Flowers (Nilsback & Zisserman, 2008) | 102 | 2,040 | 6,149 |
| Flowers(Test) (Nilsback & Zisserman, 2008) | 102 | 6,149 | 2,040 |
| Caltech101 (Fei-Fei et al., 2004) | 102 | 3,060 | 6,084 |
| OxfordPets (Parkhi et al., 2012) | 37 | 3,680 | 3,669 |
| DTD (Cimpoi et al., 2014) | 47 | 3,760 | 1,880 |
| STL-10 (Coates et al., 2011) | 10 | 5,000 | 8,000 |
| KITTI Distance (Geiger et al., 2012) | 4 | 5,985 | 1,496 |
| USPS (Sankaranarayanan et al., 2018) | 10 | 7,291 | 2,007 |
| ImageNet-10 (Chang et al., 2017) | 10 | 10,500 | 2,630 |
| ImageNet-Dogs (Chang et al., 2017) | 15 | 15,600 | 3,900 |
| EuroSAT (Helber et al., 2019) | 10 | 10,000 | 5,000 |
| GTSRB (Stallkamp et al., 2012) | 43 | 26,640 | 12,630 |
| FER2013 (Goodfellow et al., 2013) | 7 | 28,709 | 7,178 |
| CIFAR-10 (DeVries & Taylor, 2017) | 10 | 50,000 | 10,000 |
| CIFAR-100 (DeVries & Taylor, 2017) | 100 | 50,000 | 10,000 |
| TinyImageNet (Le & Yang, 2015) | 200 | 100,000 | 10,000 |
| Fashion-MNIST (Xiao et al., 2017) | 10 | 60,000 | 10,000 |
| Food101 (Bossard et al., 2014) | 101 | 75,750 | 25,250 |
| FGVC Aircraft (Maji et al., 2013) | 100 | 6,667 | 3,333 |
| PatchCamelyon (Veeling et al., 2018) | 2 | 294,912 | 32,768 |
| UCF101 (Soomro et al., 2012) | 101 | 9,537 | 3,783 |
| Country211 (Radford et al., 2021) | 211 | 42,200 | 21,100 |
| HatefulMemes (Kiela et al., 2020) | 2 | 8,500 | 500 |
| The Rendered SST2 (Radford et al., 2021) | 2 | 7,792 | 1,821 |
| Resisc45 (Cheng et al., 2017) | 45 | 25,200 | 6,300 |

## A.2 FOUNDATION REPRESENTATIONS

SEAL is compatible with many pre-trained representations. This paper presents a comprehensive evaluation of SEAL on a wide range of representation spaces that vary in pre-training datasets, model architectures, and training objectives. Specifically, for 1-space, we consider CLIP ResNets (ResNet50, ResNet101, ResNet50x4), CLIP Vision Transformers (VIT-B/32, VIT-B/16, VIT-L/14) pre-trained on WebImageText-400M (Radford et al., 2021), as well as DINOv2 VIT-G/14 pre-trained on LVD-142M (Oquab et al., 2023). SEAL combines two representation spaces to form a 2-space. The specific backbones used in each experiment are indicated in the corresponding sections. For all models, representations are precomputed using standard image preprocessing pipelines. Details of the pre-trained representations are provided in Table 3.

Table 3: Foundation representations, including architecture, number of parameters, and textual supervision status.

| Model | Architecture | Parameters | Trained on | Textual |
|---|---|---|---|---|
| CLIP(Radford et al., 2021) | RN50 | 100M | WebImageText-400M | ✓ |
| | RN101 | 120M | | |
| | RN50x4 | 180M | | |
| | VIT-B/32 | 150M | | |
| | VIT-B/16 | 150M | | |
| | VIT-L/14 | 430M | | |
| DINOv2 (Oquab et al., 2023) | VIT-G/14 | 1.1B | LVD-142M | ✕ |

## B  EFFECT OF MUTUAL DISTILLATION

### B.1  RESULTS OF 2-SPACE BACKBONES ON THE 26 DATASETS

To explore the benefits of mutual distillation in a 2-space setting, we evaluate SEAL on two representative 2-space backbone combinations: CLIP VIT-B/32 + CLIP VIT-B/16 and CLIP VIT-L/14 + DINOv2. Figure 8 presents the performance of SEAL (with Mutual Distillation) compared to the baseline TURTLE (without Mutual Distillation) in 26 datasets using radar charts.

For CLIP VIT-B/32 + CLIP VIT-B/16, SEAL consistently outperforms the baseline TURTLE model on most datasets. Notable improvements include datasets such as OxfordPets (+43.4%), ImageNet-Dogs (+41.41%), and EuroSAT (+10.84%), indicating that Mutual Distillation effectively improves the structural fidelity of embeddings. The radar chart illustrates that the combined representation provides more robust performance across datasets.

Similarly, for CLIP VIT-L/14 + DINOv2, SEAL demonstrates clear performance gains over TUR-TLE, with pronounced improvements on datasets such as GTSRB (+7.9%) and USPS(+14.3%).

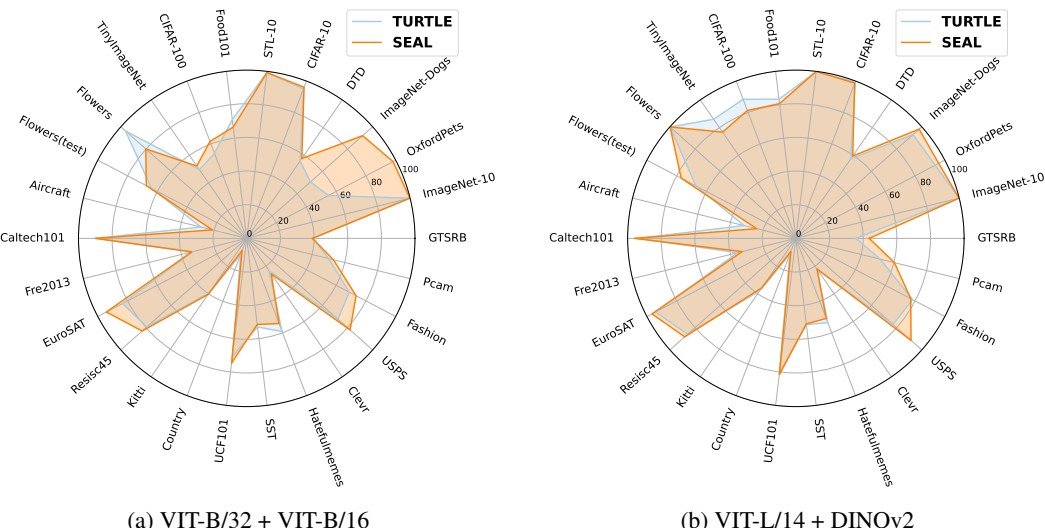

(a) VIT-B/32 + VIT-B/16                    (b) VIT-L/14 + DINOv2

Figure 8: Effect of Mutual Distillation on 2-space backbones. The radar charts show the accuracy (%) of the TURTLE baseline (represents without Mutual Distillation) and SEAL (represents with Mutual Distillation) across 26 datasets.

### B.2  RESULTS OF 1-SPACE BACKBONE ON 26 DATASETS

To evaluate the effect of SEAL on 1-space backbone, we conducted experiments on the 26 datasets. For each dataset, We measured the clustering accuracy (ACC) of SEAL (with Mutual Distillation) and the TURTLE (without Mutual Distillation) baseline across seven representative backbones, which are denoted as 1–7 in the Figure 9: (1) CLIP ResNet50, (2) CLIP ResNet101, (3) CLIP ResNet50×4, (4) CLIP ViT-B/32, (5) CLIP ViT-B/16, (6) DINOv2, and (7) CLIP ViT-L/14.

Figure 9 shows a two-panel visualization for each dataset. The upper panel displays the ACC gain (*SEAL minus TURTLE*) for each backbone, highlighting the improvement introduced by SEAL. Points above the zero line correspond to improvements by SEAL, whereas points below indicate a decrease in performance. The lower panel directly compares the ACC of TURTLE and SEAL on per backbone. Across most of the datasets, SEAL outperforms TURTLE, achieving the largest gains on ResNet-based backbones and moderate but noticeable improvements on Transformer backbones. These results demonstrate that SEAL effectively enhances 1-space visual representations, yielding consistent and interpretable gains in clustering performance.

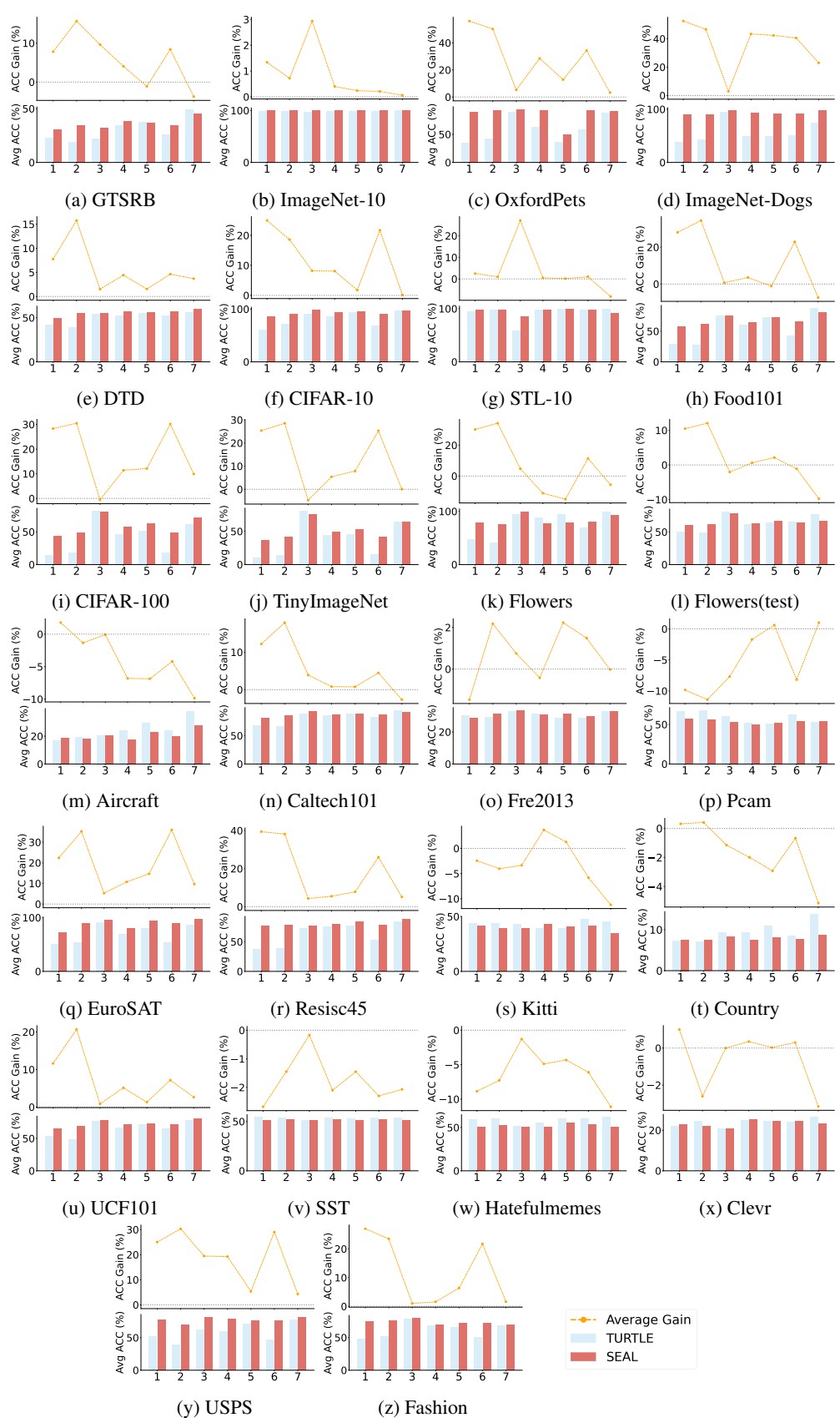

Figure 9: Effect of Mutual Distillation on 1-space backbone for each data dataset. SEAL leverages Mutual Distillation, while TURTLE serves as the counterpart without it.

## C    CLUSTERING STABILITY ACROSS 7 BACKBONES ON 26 DATASETS

Figure 10 presents a comprehensive view of the stability of clustering in the 26 datasets considered in our experiments. For each dataset, we evaluated the clustering accuracy distribution across 7 different backbones, comparing the performance of TURTLE and SEAL.

Overall, SEAL consistently demonstrates higher stability than TURTLE in most datasets. On datasets with simpler class structures, such as CIFAR-10, STL-10, and GTSRB, both methods achieve relatively high and stable clustering accuracy, but SEAL exhibits slightly narrower variance, indicating more reliable performance across backbones.

For more complex datasets, including ImageNet-10, OxfordPets, TinyImageNet, and Flowers, the advantage of SEAL becomes more pronounced. TURTLE's clustering results show larger variability between different backbones, whereas SEAL maintains higher median accuracy with reduced spread, highlighting its robustness to the choice of representation.

Remote sensing datasets such as EuroSAT and Resisc45, as well as specialized datasets like Aircraft and Pcam, also benefit from SEAL's Mutual Distillation, showing a more concentrated distribution of clustering accuracy. This indicates that SEAL can better capture fine-grained structural cues, leading to consistent clustering performance.

Finally, for more challenging datasets, such as UCF101, SEAL not only improves the median clustering accuracy but also significantly reduces the variance across backbones, demonstrating its ability to generalize well across diverse visual domains.

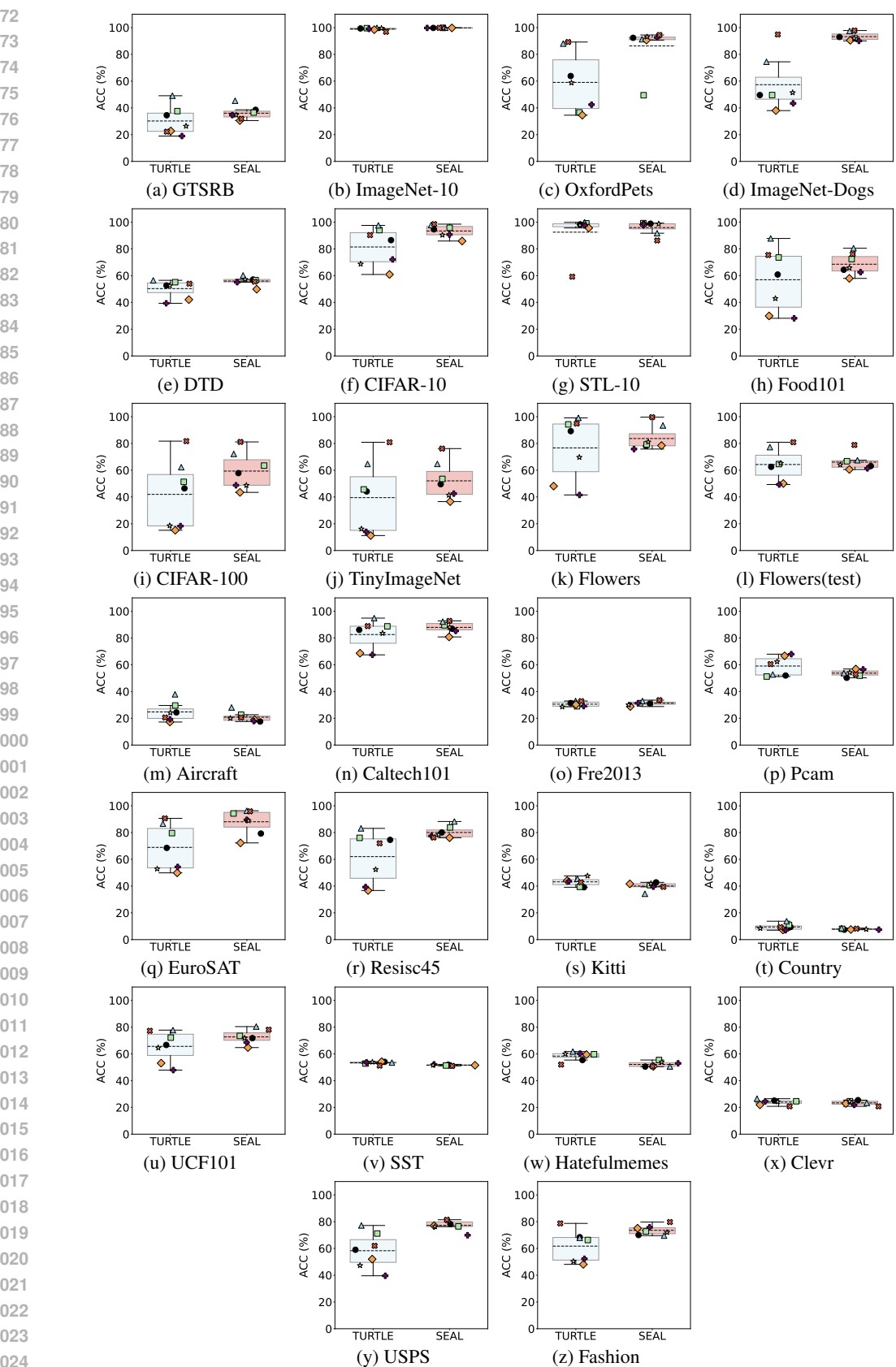

Figure 10: Detailed clustering stability plots for all 26 datasets. Each subplot shows the distribution of clustering accuracy across 7 backbones for Turtle and SEAL.

## D    SEPARABILITY IMPROVEMENT OF SPATIAL EMBEDDING

We evaluate the separability of representations by triplet accuracy. Triplet accuracy is computed based on the standard triplet definition: for a given anchor image, a positive image (same class) and a negative image (different class) are sampled. A triplet is considered correct if the distance between the anchor and positive embeddings is smaller than the distance between the anchor and negative embeddings. Formally, consider a triplet $(x_i, x_j, x_k)$, where $x_i$ and $x_j$ belong to the same class, and $x_i$ and $x_k$ belong to different classes. Let the corresponding embeddings be $f(x_i), f(x_j), f(x_k)$. The triplet is regarded as *correct* if

$$\text{dist}\big(f(x_i), f(x_j)\big) < \text{dist}\big(f(x_i), f(x_k)\big),$$

where $\text{dist}(\cdot, \cdot)$ denotes a distance function between two embeddings. In this experiment, we utilize cosine distance.

Table 4 reports the comparison of triplet verification accuracy on 10,000 randomly sampled images across 26 datasets before (*Original, the Visual Embedding*) and after (*Enhanced, the Spatially Aware Embedding*) applying mutual distillation. The results are evaluated on multiple backbone networks, including CLIP ResNets (ResNet50, ResNet101, ResNet50*4), CLIP Vision Transformers (VIT-B/32, VIT-B/16, VIT-L/14), and DINOv2.

Overall, spatially aware embedding improves triplet accuracy across almost all datasets and backbone architectures. Notable gains are observed in fine-grained datasets with high intra-class variation, such as *OxfordPets* and *ImageNet-Dogs*, demonstrating the effectiveness of incorporating spatial structure information into the foundation model embedding. On average, the triplet accuracy increases by approximately 5–6% across different backbones, with DINOv2 achieving the highest post-distillation performance.

Since higher triplet accuracy indicates that semantically similar images are closer in the embedding space while dissimilar images are farther apart, the improvement in triplet accuracy directly contributes to more separable feature clusters. Therefore, spatially aware embedding not only improves the linear separability of foundation model embeddings, making different-class data easier to distinguish, but also benefits downstream clustering performance, enabling more accurate and stable grouping of images.

## E    PERFORMANCE COMPARISON BETWEEN K-MEANS 1-SPACE, TURTLE 1-SPACE, AND SEAL 1-SPACE

Table 5 presents a comprehensive comparison of clustering ACC across the 26 datasets using three clustering approaches: K-Means 1-space, TURTLE 1-space, and SEAL 1-space. For each method, we evaluate multiple pre-trained backbones, including CLIP ResNets (50, 101, 50x4), CLIP VITs (B/32, B/16, L/14) and DINOV2.

In K-Means 1-space, the image features extracted from each backbone are directly clustered using standard K-Means without additional processing. TURTLE 1-space improves on this by discovering the underlying human labeling within the feature space. SEAL 1-space further enhances performance by jointly discovering discriminative features and the underlying human labeling, effectively leveraging both spatial structure and semantic cues.

Overall, SEAL 1-space consistently achieves higher average accuracy compared to TURTLE 1-space and K-Means 1-space. The improvement is particularly notable on datasets with fine-grained categories, such as OxfordPets, ImageNet-Dogs, and Flowers, where SEAL 1-space maintains robust performance across different backbones.

These results indicate that SEAL captures richer information, producing more discriminative representations in a 1-space setting. This demonstrates the effectiveness of incorporating spatial and semantic cues for robust clustering across diverse datasets and backbones.

| | ClipResNet50 | | ClipResNet101 | | ClipResNet50*4 | | ClipVITB/32 | | ClipVITB/16 | | DINOv2 | | ClipVITL/14 | |
|---|---|---|---|---|---|---|---|---|---|---|---|---|---|---|
| | Origin | Enhance | Origin | Enhance | Origin | Enhance | Origin | Enhance | Origin | Enhance | Origin | Enhance | Origin | Enhance |
| GTSRB | 75.62 | 83.99 | 77.14 | 85.80 | 76.71 | 86.43 | 74.64 | 82.43 | 77.94 | 88.38 | 69.13 | 82.91 | 84.79 | 92.90 |
| ImageNet-10 | 97.13 | 99.95 | 97.30 | 99.93 | 97.23 | 99.94 | 97.35 | 99.92 | 97.36 | 99.95 | 96.46 | 99.92 | 96.86 | 99.95 |
| OxfordPets | 87.93 | 98.91 | 91.11 | 99.10 | 91.20 | 99.40 | 88.37 | 99.35 | 65.14 | 84.21 | 83.23 | 95.65 | 93.29 | 99.65 |
| ImageNet-Dogs | 69.74 | 95.94 | 75.42 | 97.28 | 76.50 | 97.62 | 71.26 | 97.01 | 73.98 | 97.30 | 97.75 | 99.42 | 78.08 | 99.05 |
| DTD | 80.48 | 89.36 | 81.22 | 90.11 | 82.42 | 90.40 | 81.04 | 89.65 | 81.97 | 90.27 | 86.33 | 91.06 | 81.25 | 91.20 |
| CIFAR-10 | 81.00 | 92.87 | 85.32 | 95.77 | 84.67 | 95.66 | 87.97 | 97.60 | 87.93 | 98.45 | 89.54 | 99.56 | 86.12 | 98.93 |
| STL-10 | 94.26 | 99.42 | 94.84 | 99.70 | 95.06 | 99.80 | 94.62 | 99.80 | 94.28 | 99.74 | 83.48 | 97.84 | 92.14 | 99.32 |
| Food101 | 89.11 | 92.65 | 90.65 | 93.85 | 91.50 | 94.43 | 89.70 | 93.93 | 92.67 | 95.55 | 96.95 | 97.18 | 94.15 | 97.50 |
| CIFAR-100 | 80.07 | 90.64 | 84.83 | 92.61 | 83.67 | 92.47 | 86.82 | 95.43 | 86.44 | 96.03 | 96.01 | 98.66 | 84.74 | 97.31 |
| TinyImageNet | 84.00 | 88.25 | 86.94 | 90.15 | 86.27 | 89.75 | 90.68 | 93.15 | 90.47 | 93.59 | 97.20 | 95.81 | 89.77 | 94.55 |
| Flowers | 94.66 | 97.21 | 95.59 | 96.52 | 98.04 | 97.35 | 96.81 | 96.91 | 97.40 | 97.75 | 99.85 | 99.95 | 98.19 | 99.46 |
| Flowers(test) | 94.67 | 97.66 | 96.05 | 98.10 | 96.94 | 98.46 | 96.28 | 98.32 | 97.32 | 98.81 | 99.89 | 99.85 | 98.55 | 99.50 |
| Aircraft | 79.78 | 77.67 | 82.30 | 76.45 | 84.58 | 78.27 | 83.05 | 76.59 | 86.68 | 79.27 | 87.48 | 86.40 | 89.25 | 83.04 |
| Caltech101 | 95.49 | 98.46 | 96.86 | 99.22 | 97.22 | 98.99 | 97.09 | 99.15 | 97.25 | 99.31 | 98.99 | 99.61 | 97.48 | 99.71 |
| Fre2013 | 56.27 | 59.59 | 57.80 | 58.91 | 57.93 | 59.77 | 57.96 | 59.18 | 58.05 | 59.39 | 61.16 | 61.42 | 57.56 | 61.45 |
| Pcam | 57.86 | 54.19 | 58.29 | 53.76 | 55.05 | 53.31 | 56.03 | 52.94 | 54.53 | 53.28 | 55.41 | 62.16 | 56.64 | 53.90 |
| EuroSAT | 80.51 | 94.15 | 81.18 | 96.27 | 82.89 | 96.03 | 81.64 | 94.56 | 84.26 | 97.79 | 84.09 | 98.01 | 87.86 | 98.77 |
| Resisc45 | 91.18 | 96.63 | 93.02 | 97.49 | 92.89 | 97.24 | 93.55 | 98.01 | 95.20 | 98.30 | 92.38 | 97.35 | 95.45 | 98.62 |
| Kitti | 59.60 | 60.12 | 58.10 | 55.81 | 59.13 | 60.05 | 58.48 | 60.13 | 58.75 | 58.96 | 59.85 | 59.20 | 58.96 | 56.91 |
| Country | 64.23 | 61.03 | 64.73 | 60.17 | 66.39 | 60.61 | 63.64 | 59.84 | 65.58 | 61.52 | 62.55 | 59.53 | 67.83 | 62.40 |
| UCF101 | 92.70 | 96.52 | 94.63 | 96.69 | 94.24 | 97.51 | 94.57 | 97.33 | 95.88 | 97.49 | 98.22 | 98.40 | 97.55 | 99.02 |
| SST | 50.87 | 49.45 | 51.63 | 51.49 | 52.09 | 51.22 | 50.73 | 51.57 | 51.04 | 51.37 | 50.73 | 50.54 | 52.35 | 51.59 |
| Hatefulmemes | 51.89 | 49.81 | 51.07 | 50.88 | 50.80 | 51.47 | 49.61 | 49.75 | 49.54 | 51.66 | 50.14 | 50.59 | 50.95 | 50.59 |
| Clevr | 60.10 | 60.10 | 62.00 | 58.90 | 58.75 | 60.75 | 59.90 | 61.35 | 58.70 | 60.55 | 55.45 | 54.80 | 59.20 | 58.95 |
| USPS | 85.01 | 96.23 | 81.17 | 96.80 | 82.49 | 96.54 | 87.30 | 97.83 | 87.97 | 97.52 | 83.23 | 95.65 | 88.70 | 97.81 |
| Fashion | 81.12 | 90.72 | 84.10 | 91.80 | 82.20 | 91.88 | 84.82 | 92.48 | 82.64 | 91.58 | 87.48 | 93.94 | 81.66 | 91.64 |
| **AVG.** | 78.28 | 83.52 | 79.74 | 83.98 | 79.88 | 84.77 | 79.77 | 84.39 | 79.61 | 84.54 | 81.65 | 85.59 | 81.51 | 85.39 |

Table 4: Triplet accuracy (%) on 10,000 randomly-sampled images before (Origin) and after (Enhance) mutual distillation. Underlined numbers indicate the higher value within each Origin/Enhance pair for the same architecture.

| | ClipVIT-B/32 | | | ClipVIT-B/16 | | | ClipVIT-L/14 | | | ClipResNet50 | | | ClipResNet101 | | | ClipResNet50*4 | | | DINOV2 | | |
|---|---|---|---|---|---|---|---|---|---|---|---|---|---|---|---|---|---|---|---|---|---|
| | K-Means | TURTLE | SEAL | K-Means | TURTLE | SEAL | K-Means | TURTLE | SEAL | K-Means | TURTLE | SEAL | K-Means | TURTLE | SEAL | K-Means | TURTLE | SEAL | K-Means | TURTLE | SEAL |
| GTSRB | 33.64 | 34.50 | 38.56 | 35.54 | 37.59 | 36.49 | 49.06 | 49.07 | 45.33 | 26.66 | 22.73 | 30.52 | 28.69 | 18.95 | 34.62 | 26.10 | 26.37 | 34.77 | 20.71 | 22.22 | 31.85 |
| ImageNet-10 | 98.69 | 99.36 | 99.76 | 99.22 | 99.58 | 99.82 | 99.63 | 99.80 | 99.86 | 96.97 | 98.43 | 99.77 | 98.46 | 99.02 | 99.74 | 98.93 | 99.53 | 99.74 | 91.84 | 96.86 | 99.80 |
| OxfordPets | 52.61 | 63.83 | 92.36 | 30.38 | 36.63 | 49.46 | 69.97 | 88.02 | 91.30 | 47.77 | 34.54 | 90.65 | 57.77 | 42.42 | 92.64 | 60.57 | 58.75 | 93.10 | 82.23 | 89.21 | 94.48 |
| ImageNet-Dogs | 39.80 | 49.58 | 92.97 | 42.57 | 49.58 | 91.94 | 62.99 | 74.43 | 97.44 | 33.25 | 37.97 | 90.51 | 46.47 | 43.39 | 90.08 | 47.12 | 51.38 | 91.96 | 87.52 | 94.86 | 97.81 |
| DTD | 46.38 | 52.70 | 57.13 | 47.77 | 55.13 | 56.68 | 50.88 | 56.52 | 60.21 | 39.65 | 42.10 | 49.87 | 41.86 | 39.36 | 55.13 | 45.88 | 52.58 | 57.21 | 47.55 | 53.99 | 55.51 |
| CIFAR-10 | 76.11 | 86.55 | 94.63 | 78.31 | 94.04 | 95.81 | 83.54 | 97.58 | 97.76 | 55.27 | 60.90 | 85.90 | 69.52 | 72.12 | 90.71 | 65.36 | 68.70 | 90.42 | 74.92 | 90.32 | 98.51 |
| STL-10 | 97.62 | 98.36 | 99.88 | 94.84 | 99.12 | 99.26 | 95.74 | 99.86 | 91.72 | 88.94 | 95.66 | 98.16 | 96.70 | 97.58 | 98.56 | 97.02 | 97.72 | 98.72 | 53.12 | 59.22 | 86.32 |
| Food101 | 56.13 | 60.86 | 64.41 | 70.64 | 73.59 | 72.42 | 79.80 | 87.82 | 80.52 | 66.91 | 29.93 | 57.97 | 70.45 | 28.16 | 62.62 | 70.97 | 42.98 | 65.88 | 70.65 | 75.43 | 76.12 |
| CIFAR-100 | 42.39 | 46.44 | 57.85 | 47.82 | 51.31 | 63.46 | 53.64 | 62.27 | 72.18 | 25.04 | 15.13 | 43.46 | 34.99 | 18.25 | 48.74 | 30.36 | 18.56 | 48.74 | 68.47 | 80.90 | 81.16 |
| TinyImageNet | 40.85 | 44.15 | 49.56 | 43.37 | 45.65 | 53.52 | 57.49 | 64.66 | 67.47 | 25.22 | 11.17 | 36.54 | 34.40 | 13.97 | 42.51 | 34.72 | 16.11 | 41.42 | 72.19 | 80.91 | 76.17 |
| Flowers | 70.64 | 89.22 | 78.09 | 81.27 | 94.26 | 79.26 | 88.63 | 99.12 | 93.43 | 67.50 | 48.14 | 78.43 | 69.31 | 41.52 | 75.83 | 80.00 | 69.66 | 81.13 | 96.18 | 94.85 | 99.66 |
| Flowers(test) | 70.64 | 62.53 | 63.13 | 81.27 | 64.60 | 66.73 | 88.63 | 77.25 | 67.47 | 67.50 | 50.07 | 60.56 | 69.31 | 49.33 | 61.38 | 80.00 | 65.21 | 64.09 | 96.18 | 71.98 | 78.86 |
| Aircraft | 22.80 | 24.40 | 17.59 | 27.22 | 29.58 | 22.71 | 33.57 | 37.93 | 28.08 | 20.25 | 17.22 | 19.00 | 23.02 | 19.30 | 17.97 | 24.31 | 24.22 | 20.02 | 18.87 | 20.58 | 20.50 |
| Caltech101 | 72.29 | 86.11 | 86.93 | 76.99 | 88.73 | 89.51 | 84.51 | 94.87 | 92.22 | 62.06 | 68.63 | 80.82 | 69.22 | 67.42 | 85.29 | 73.24 | 83.56 | 88.04 | 85.82 | 88.89 | 92.81 |
| Fer2013 | 26.95 | 31.38 | 30.96 | 27.82 | 28.99 | 31.22 | 31.48 | 32.93 | 32.91 | 26.42 | 30.28 | 28.81 | 26.79 | 29.12 | 31.30 | 27.65 | 28.58 | 30.07 | 32.98 | 32.81 | 33.56 |
| Pcam | 62.05 | 51.97 | 50.27 | 62.24 | 51.22 | 51.83 | 64.30 | 52.73 | 53.73 | 64.30 | 66.68 | 56.84 | 64.28 | 67.93 | 56.53 | 62.01 | 62.36 | 54.19 | 59.43 | 60.50 | 52.83 |
| EuroSAT | 64.14 | 68.55 | 79.28 | 73.98 | 79.59 | 94.27 | 74.99 | 86.60 | 96.29 | 56.83 | 49.89 | 72.27 | 52.31 | 54.31 | 89.54 | 53.39 | 52.93 | 88.89 | 62.36 | 90.61 | 95.89 |
| Resisc45 | 68.19 | 74.56 | 80.07 | 73.36 | 76.05 | 83.83 | 73.87 | 83.14 | 88.27 | 54.29 | 36.72 | 76.12 | 62.56 | 39.27 | 77.45 | 66.54 | 52.44 | 78.43 | 62.65 | 71.98 | 76.30 |
| Kitti | 48.09 | 39.13 | 42.79 | 48.89 | 39.43 | 40.70 | 48.35 | 45.45 | 34.25 | 47.25 | 44.11 | 41.67 | 48.32 | 39.55 | 39.55 | 47.95 | 41.82 | 41.82 | 49.16 | 42.79 | 39.43 |
| Country | 9.51 | 9.38 | 7.39 | 10.57 | 10.98 | 8.06 | 12.99 | 13.74 | 8.60 | 8.64 | 7.22 | 7.54 | 9.37 | 7.03 | 7.45 | 9.87 | 8.39 | 7.72 | 9.23 | 9.34 | 8.19 |
| UCF101 | 61.34 | 66.66 | 71.79 | 63.53 | 72.15 | 73.44 | 70.94 | 77.75 | 80.38 | 51.87 | 53.09 | 64.71 | 59.64 | 47.89 | 68.59 | 61.85 | 64.54 | 71.72 | 71.33 | 77.26 | 78.10 |
| SST | 52.21 | 54.03 | 51.93 | 54.52 | 52.77 | 51.32 | 51.73 | 53.49 | 51.42 | 51.75 | 52.13 | 51.41 | 53.80 | 53.62 | 52.17 | 55.25 | 53.93 | 53.95 | 51.16 | 51.21 | 51.04 |
| Hatefulmemes | 60.15 | 55.38 | 50.49 | 60.91 | 59.74 | 55.42 | 62.72 | 61.71 | 50.60 | 60.85 | 59.48 | 50.62 | 60.59 | 60.20 | 52.93 | 60.86 | 60.05 | 54.19 | 55.60 | 52.00 | 50.72 |
| Clevr | 24.95 | 25.10 | 25.45 | 24.30 | 24.60 | 24.63 | 23.50 | 26.50 | 23.35 | 24.05 | 21.90 | 22.90 | 26.05 | 24.50 | 21.90 | 25.30 | 24.25 | 24.55 | 18.70 | 20.80 | 20.80 |
| USPS | 60.76 | 59.00 | 78.23 | 78.47 | 71.13 | 76.49 | 79.66 | 77.16 | 81.44 | 61.42 | 52.00 | 77.04 | 51.28 | 39.57 | 69.87 | 51.83 | 47.25 | 76.30 | 56.38 | 61.97 | 81.42 |
| Fashion | 64.16 | 68.43 | 70.01 | 58.96 | 66.31 | 72.69 | 53.19 | 67.93 | 69.56 | 58.78 | 48.10 | 75.09 | 63.83 | 52.27 | 75.77 | 56.05 | 50.29 | 72.00 | 65.37 | 78.71 | 79.72 |
| **AVG.** | 54.75 | 57.78 | 62.71 | 57.23 | 59.71 | 63.11 | 62.79 | 68.01 | 67.42 | 48.64 | 44.47 | 59.51 | 52.71 | 45.00 | 61.50 | 53.43 | 51.08 | 62.56 | 59.93 | 64.61 | 67.60 |

Table 5: Clustering accuracy comparison across 7 backbones ( ClipResNet50, ClipResNet101, ClipResNet50*4, ClipVIT-B/32, ClipVIT-B/16, ClipVIT-L/14, DINOV2). For each backbone, results are reported under K-Means 1-space, TURTLE 1-space, and SEAL 1-space. The Underlined value indicates the best-performing method within that backbone.

# F  PERFORMANCE COMPARISON BETWEEN K-MEANS 2-SPACE, TURTLE 2-SPACE, AND SEAL 2- SPACE

Table 6 presents a detailed comparison of clustering accuracy across 26 diverse vision datasets using three 2-space clustering approaches: K-Means 2-space, TURTLE 2-space, and SEAL 2-space. Each method is evaluated with different backbone combinations, including CLIP VIT-L/14 + DINOv2 and CLIP VIT-B/32 + CLIP VIT-B/16.

Overall, SEAL 2-space achieves the highest average accuracy compared to TURTLE 2-space and K-Means 2-space. The performance gains are particularly notable on datasets with complex structures categories, such as EuroSAT, CIFAR-10, OxfordPets, and ImageNet-Dogs, where SEAL 2-space maintains robust performance across different combinations.

In addition, we report the clustering performance of K-means on the 26 datasets in two settings, which are (1) directly concatenating the CLIP VIT-L/14 and DINOv2 embeddings and (2) applying the mutual distillation to both embeddings before concatenation. The results are listed in Table 7. From Table 7, it can be seen that spatially aware embeddings significantly improve performance on most of the datasets, which illustrates the effectiveness of mutual distillation.

These results demonstrate that jointly leveraging two embedding spaces and spatial structure cues enables SEAL to produce more discriminative and semantically meaningful representations, leading to superior clustering performance compared to standard K-Means and TURTLE in a 2-space setting.

| | VIT-B/32 / VIT-B/16 | | | VIT-L/14 / DINOv2 | | |
|---|---|---|---|---|---|---|
| | K-Means | TURTLE | SEAL | K-Means | TURTLE | SEAL |
| GTSRB | 36.25 | 39.33 | 39.09 | 22.81 | 35.73 | 43.61 |
| ImageNet-10 | 99.18 | 99.62 | 99.82 | 92.67 | 99.79 | 99.87 |
| OxfordPets | 52.69 | 54.81 | 98.18 | 83.10 | 93.54 | 95.87 |
| ImageNet-Dogs | 42.30 | 50.62 | 92.03 | 86.37 | 93.03 | 97.87 |
| DTD | 49.04 | 55.85 | 57.61 | 48.88 | 57.45 | 59.44 |
| CIFAR-10 | 79.14 | 95.01 | 96.07 | 85.00 | 99.47 | 98.69 |
| STL-10 | 98.34 | 99.22 | 99.26 | 51.54 | 99.74 | 99.94 |
| Food101 | 66.94 | 71.08 | 66.63 | 72.08 | 83.31 | 80.63 |
| CIFAR-100 | 48.60 | 54.08 | 61.59 | 69.72 | 88.38 | 81.28 |
| TinyImageNet | 44.65 | 49.55 | 52.27 | 74.70 | 86.02 | 76.70 |
| Flowers | 82.35 | 96.27 | 80.15 | 98.68 | 99.80 | 99.85 |
| Flowers(test) | 74.58 | 63.95 | 67.43 | 93.71 | 67.12 | 77.00 |
| Aircraft | 26.73 | 28.09 | 21.06 | 18.75 | 31.66 | 23.98 |
| Caltech101 | 75.33 | 89.77 | 89.87 | 85.46 | 93.86 | 95.88 |
| Fre2013 | 27.74 | 32.87 | 33.76 | 33.10 | 35.76 | 32.89 |
| Pcam | 67.20 | 50.03 | 53.00 | 58.76 | 51.67 | 60.33 |
| EuroSAT | 71.85 | 83.37 | 94.21 | 62.72 | 94.20 | 96.60 |
| Resisc45 | 49.06 | 79.38 | 83.04 | 67.38 | 85.90 | 88.27 |
| Kitti | 9.95 | 39.31 | 40.47 | 49.16 | 37.94 | 35.69 |
| Country | 65.32 | 10.21 | 7.84 | 9.28 | 10.25 | 8.36 |
| UCF101 | 54.94 | 70.67 | 74.57 | 72.47 | 81.81 | 81.38 |
| SST | 60.62 | 53.04 | 51.55 | 51.17 | 51.66 | 51.23 |
| Hatefulmemes | 24.95 | 59.25 | 54.13 | 55.67 | 53.59 | 50.89 |
| Clevr | 82.47 | 23.45 | 25.65 | 20.05 | 22.10 | 22.10 |
| USPS | 65.67 | 72.42 | 82.03 | 57.66 | 77.11 | 91.43 |
| Fashion | 62.71 | 68.81 | 73.40 | 65.38 | 78.33 | 77.09 |
| **AVG.** | 58.41 | 61.16 | 65.18 | 61.01 | 69.59 | 70.26 |

Table 6: Clustering accuracy comparison across different backbone combination settings and methods. The Underlined value indicates the best-performing method within that combination setting.

| | K-Means Clustering on | |
|---|---|---|
| Dataset | Directly Concatenating VIT-L/14 and DINOv2 | Spatially Aware Embedding with VIT-L/14 and DINOv2 |
| GTSRB | 22.81 | 41.14 |
| ImageNet-10 | 92.67 | 99.79 |
| OxfordPets | 83.10 | 90.41 |
| ImageNet-Dogs | 86.37 | 97.66 |
| DTD | 48.88 | 60.00 |
| CIFAR-10 | 85.00 | 98.66 |
| STL-10 | 51.54 | 99.56 |
| Food101 | 72.08 | 75.51 |
| CIFAR-100 | 69.72 | 72.57 |
| TinyImageNet | 74.70 | 70.54 |
| Flowers | 98.68 | 94.90 |
| Flowers(test) | 93.71 | 91.38 |
| Aircraft | 18.75 | 21.84 |
| Caltech101 | 85.46 | 88.27 |
| Fre2013 | 33.10 | 33.85 |
| Pcam | 58.76 | 60.21 |
| EuroSAT | 62.72 | 96.26 |
| Resisc45 | 67.38 | 79.03 |
| Kitti | 49.16 | 46.68 |
| Country | 9.28 | 8.18 |
| UCF101 | 72.47 | 78.83 |
| SST | 51.17 | 51.18 |
| Hatefulmemes | 55.67 | 50.73 |
| Clevr | 20.05 | 20.35 |
| USPS | 57.66 | 82.79 |
| Fashion | 65.38 | 71.48 |
| **AVG.** | 61.01 | 68.53 |

Table 7: K-Means accuracy from concatenating embeddings of VIT-L/14 and DINOv2 after applying a spatially-aware transformation to each, compared to directly concatenating original embeddings of VIT-L/14 and DINOv2, on 26 benchmark datasets.

