# OpenReview forum: "Unsupervised learning with spatial embedding and human labeling"
_ICLR.cc/2026/Conference — ICLR 2026 Conference Withdrawn Submission_

### Official Review · Reviewer_hBBy · 2025-10-27

**Soundness:** 3
**Presentation:** 3
**Contribution:** 2
**Rating:** 4
**Confidence:** 3

**Summary:**

This paper proposes SEAL, an unsupervised learning framework for clustering. SEAL enhances the feature separability of large-scale pretrained vision models (e.g., CLIP, DINOv2) by integrating spatial embeddings extracted from a Graph Attention Network (GAT) with the original foundation model features.

The spatial embeddings capture relational cues among image patches, which are fused with the foundation model’s features via mutual distillation to produce spatially aware embeddings.
A lightweight linear classifier is then trained on this embedding space to infer human-consistent cluster assignments, thereby recovering semantic labeling without supervision.
Extensive experiments on 26 benchmark datasets and 7 backbone models demonstrate that SEAL significantly improves clustering performance and feature separability compared to prior state-of-the-art unsupervised methods such as TURTLE.

**Strengths:**

1. The paper proposes an integration of spatial structure into unsupervised learning. The use of GAT-based spatial embeddings adds an interpretable and principled way to capture spatial dependencies between image regions, which is rarely explored in unsupervised clustering.

2. The mutual distillation strategy is well-designed and effective. By aligning visual and spatial modalities, the approach enhances embedding separability while maintaining robustness, which is validated through extensive empirical analysis.

3. Extensive experiments provide strong evidence for the method’s effectiveness. The authors evaluate SEAL on 26 datasets and 7 backbone architectures, covering a wide range of domains, demonstrating consistent improvement and generalization ability.

4. Quantitative and qualitative metrics are comprehensive. The paper uses ACC, NMI, ARI, and triplet accuracy to rigorously evaluate performance, along with ablation studies and parameter sensitivity analysis.

**Weaknesses:**

1. The novelty relative to TURTLE is somewhat incremental. Although the introduction of spatial embeddings is meaningful, much of the overall framework—including the bi-level optimization and human labeling concept—is inherited from prior work (TURTLE).

2. The theoretical foundation for mutual distillation could be strengthened. The paper provides empirical justification for separability improvement but lacks a deeper theoretical analysis explaining why spatial–visual fusion leads to more linearly separable representations.

3. Qualitative visualizations are limited. The paper would benefit from more examples of embedding visualizations or clustering maps to illustrate how spatial awareness improves semantic grouping.

4. Generality beyond vision is only briefly mentioned. Although the authors suggest possible extension to multimodal data, there is no experimental evidence or discussion about cross-modal robustness.

5. The “human labeling” terminology could be misleading. Since no actual human annotations are used, the phrase might suggest a level of semantic alignment that the model does not fully achieve.

**Questions:**

Please address the concerns I raised in the Weaknesses section.

**Details Of Ethics Concerns:**

None.

---

### Official Review · Reviewer_MPpV · 2025-10-31

**Soundness:** 3
**Presentation:** 2
**Contribution:** 3
**Rating:** 4
**Confidence:** 5

**Summary:**

This paper proposes SEAL (Spatial Embedding and human labeling), a unsupervised visual learning method that enhances the linear separability of pretrained foundation models' visual embeddings (e.g., CLIP, DINOv2) by integrating spatial information via a Graph Attention Network (GAT) and mutual distillation between spatial and visual embeddings. A lightweight linear classifier is then trained to recover human labeling, only given the number of categories.

**Strengths:**

1) The paper addresses a key limitation, namely that when existing large-scale foundation models are used for unsupervised learning, the visual embeddings lack sufficient linear separability, making making it challenging to identify a reliable linear
classifier in the feature space.

2) The proposed SEAL achieves state-of-the-art clustering performance across three widely used benchmark datasets

3) The spatial structure introduced by SEAL significantly accelerates human labeling recovery

4) Improvement is consistent across multiple architectures (CLIP ViT, CLIP ResNet, and DINOv2).

**Weaknesses:**

1) The GAT and mutual distillation process both rely on the representations extracted by the ResNet-50 backbone; however, the ablation study does not discuss replacing ResNet-50 with other feature extractors to determine the optimal one.

2) Fig. 3 is difficult to follow. What is the Test Encoder in Fig. 3(b) and how is the human labeling process performed in Fig. 3(d)?

3) Could the authors provide more insight into the semantic alignment between spatially aware embeddings and human concepts (e.g., via visualization)?

4) The paper does not provide sufficient analysis of why GAT is the optimal choice for modeling spatial structures, nor does it present comparisons with other spatial modeling approaches.

5) The SOTAs are quite old. Some recent methods [1-3] should be compared.

[1] Liu H, Hu P, Zhang C, et al. Interactive deep clustering via value mining. Advances in Neural Information Processing Systems, 2024, 37: 42369-42387.

[2] Geng X, Zhao S, Yu Y, et al. Personalized clustering via targeted representation learning[C]//Proceedings of the AAAI Conference on Artificial Intelligence. 2025, 39(16): 16790-16798.

[3] Palumbo E, Vandenhirtz M, Ryser A, et al. From Logits to Hierarchies: Hierarchical Clustering made Simple[C]//Forty-second International Conference on Machine Learning, 2025.

**Questions:**

1) In the setting of hierarchical classification/clustering, linear separability may not facilitate the classification/clustering. Mapping features from Euclidean space to a manifold, such as hyperbolic space, would be a better.  Why is linear separability important?

2) To get the spatial embedding extraction, the pretrained ResNet-50 is used to extract features. This means that the proposed method distill the semantics from a model which is pretrained in a supervised way to a model which is pretrained in a unsupervised way. Does this means the the proposed is not a pure unsupervised learning method as the pretrained ResNet-50 was trained with labelled images? If so, is it fair to the compared methods?

---

### Official Review · Reviewer_TU1z · 2025-11-01

**Soundness:** 2
**Presentation:** 3
**Contribution:** 2
**Rating:** 4
**Confidence:** 3

**Summary:**

The paper proposes SEAL, a new framework for unsupervised learning in computer vision that combines spatial embeddings and human labeling recovery to improve clustering and representation learning. SEAL introduces a Graph Attention Network (GAT) to capture spatial relationships between image patches and produce spatial embeddings. Besides, a mutual distillation process is proposed to fuse spatial and foundation model embeddings to enhance discriminability.
Across 26 benchmark datasets and 7 foundation backbones, SEAL demonstrates improvement over baseline models.

**Strengths:**

1. Innovative spatial fusion mechanism.
The use of a GAT to extract patch-level spatial cues adds relational understanding that foundation model embeddings often miss. This is a creative way to inject structure into unsupervised learning.

2. Mutual distillation strategy.
The cross-modal distillation between visual and spatial embeddings effectively aligns and enhances both modalities, improving feature quality and separability.

3. Comprehensive empirical validation.
The authors test SEAL on 26 datasets and multiple foundation backbones, showing consistent improvements over baselines (including TURTLE and SPICE).

**Weaknesses:**

1. Computational overhead.
Although SEAL’s efficiency is described as “comparable” to TURTLE, the added GAT and distillation steps inevitably introduce overhead. The paper acknowledges this but lacks detailed scalability analysis for larger datasets.

2. Dependence on high-quality pretrained features.
SEAL relies heavily on foundation model embeddings (e.g., CLIP, DINOv2). Its performance on domains where such embeddings are weak (e.g., medical or non-natural images) is unclear.

3. Marginal improvement.
The performance improvement over baselines such as TURTLE is marginal.

4. Limited novelty beyond fusion.
While the mutual distillation concept is well executed, it builds heavily on prior work (especially TURTLE). The primary novelty lies in integrating spatial information rather than redefining the unsupervised objective itself.

**Questions:**

None.

---

### Official Review · Reviewer_Vaiy · 2025-11-01

**Soundness:** 2
**Presentation:** 1
**Contribution:** 2
**Rating:** 2
**Confidence:** 5

**Summary:**

This paper aims to improve the clustering performance on images by incorporating the spatial embedding. Concretely, the spatial embedding is obtained by borrowing the neighborhood information using GAT. Then, the original vision embedding gathered from the pre-trained CLIP can be enhanced by doing mutual distillation between the vision and spatial embedding using their linear classifier's logits. Finally, the linear classifier's weights and the encoder's parameters are updated alternatively to achieve the clustering goal. According to the reported results using various datasets, the proposed method shows better performance compared to the reported baselines.

**Strengths:**

1) In image clustering, it is reasonable to incorporate the spatial embedding to capture the local semantic meeting better. It is also interesting to apply the mutual distillation between the vision embedding obtained from the pre-trained CLIP and the spatial embedding.

2) The contribution of the spatial-aware embedding is verified in the experiment and the overall proposed method provides better performance on various datasets using the reported accuracy metric and the included baselines.

**Weaknesses:**

1) The presentation of the methodology is very hard to follow missing a lot of details. First, the reviewer has a question on the definition of the labeling distribution. Why is it within the (K-1) dimensional probability simplex, but not K? Second, there was no clear definition on R' or w'. Are these the derivatives? Third, it is not clear why ResNet-50 was used to get input for GAT. Any specific reasons on not using other backbones? Fourth, how the edge information E is defined? Fifth, according to the definition of (4), it seems that we obtain only one g(x) for the whole dataset. Right? If so, why do we need to use g(x) with a variable x? Sixth, why does sim mean in (8) to (9). Most importantly, log(.,.) is not clearly defined. Are you using the SCAN's idea? If so, it should be log<.,.>. Right?

2) If all the guesses for the above points are correct, this work has been combining existing techniques to incorporate the spatial information, which makes the technical contribution moderate.

3) Some strong clustering baselines were not included in the comparison, e.g., SCAN, CoKe, SeCu, etc. Any reasons that they are not included? More importantly, the triplet accuracy has been used for the evaluation. However, the detailed definition is not given.

**Questions:**

All relevant questions can be found in the weakness section.

---

### Note · Authors · 2025-11-13

I have read and agree with the venue's withdrawal policy on behalf of myself and my co-authors.